# The Role of m^6^A Modifications in B-Cell Development and B-Cell-Related Diseases

**DOI:** 10.3390/ijms24054721

**Published:** 2023-03-01

**Authors:** Shuqi Wang, Huanxiang Li, Zhengxing Lian, Shoulong Deng

**Affiliations:** 1Beijing Key Laboratory for Animal Genetic Improvement, National Engineering Laboratory for Animal Breeding, Key Laboratory of Animal Genetics and Breeding of the Ministry of Agriculture, College of Animal Science and Technology, China Agricultural University, Beijing 100193, China; 2NHC Key Laboratory of Human Disease Comparative Medicine, Institute of Laboratory Animal Sciences, Chinese Academy of Medical Sciences and Comparative Medicine Center, Peking Union Medical College, Beijing 100021, China

**Keywords:** B cell, immunodeficiency, m^6^A, B-cell-related diseases

## Abstract

B cells are a class of professional antigen-presenting cells that produce antibodies to mediate humoral immune response and participate in immune regulation. m^6^A modification is the most common RNA modification in mRNA; it involves almost all aspects of RNA metabolism and can affect RNA splicing, translation, stability, etc. This review focuses on the B-cell maturation process as well as the role of three m^6^A modification-related regulators—writer, eraser, and reader—in B-cell development and B-cell-related diseases. The identification of genes and modifiers that contribute to immune deficiency may shed light on regulatory requirements for normal B-cell development and the underlying mechanism of some common diseases.

## 1. Introduction

B lymphocytes develop from hematopoietic progenitor cells in bone marrow (BM) [1,2]. Hematopoietic stem cells (HSC) differentiate into B cells via downstream pluripotent progenitors, lymphoid-induced pluripotent progenitors, common lymphoprogenitors, and B-cell precursors, thus differentiating into naive B cells expressing surface immunoglobulin [3,4]. Under the effect of the internal environment of the bone marrow, the bone marrow stem cells differentiate into pre-B cells, immature B cells, and finally mature B cells, according to the established genetic sequence. The process of immunoglobulin gene rearrangement, gene activation, transcriptional expression, and so on, finally results in the unique surface marker, the B-cell antigen receptor (BCR) [5]. Naive cells receive antigen stimulation in peripheral lymphoid organs or blood and differentiate into memory B cells and plasma cells for humoral immunity(Figure 1) [6,7].

*N*6-methyladenosine (m^6^A) is the most common, abundant, and conserved internal co-transcription modification in eukaryotic cells, especially higher eukaryotic cells. m^6^A modification helps to achieve different basic biological functions at the molecular, cellular, and physiological levels. Recent studies have shown that m^6^A RNA modification plays a crucial role in both physiological and pathological conditions, and m^6^A plays an important role in regulating immune cell function and immune response. The modification of m^6^A adds another layer of regulation to an already complex pathway of gene expression regulation in mammals. m^6^A methylation is integral to the function of innate immune responses. m6A modification controls a variety of innate immune responses, such as interferon expression, inflammatory responses, and homeostasis of macrophages and dendritic cells. However, little is known about the role of m^6^A in B-cell development and B-cell-related diseases. In this review, we summarize recent findings regarding the influence of m^6^A on B-cell development and its role in B-cell-related diseases.

## 2. B-Cell Development and Common B-Cell Immune Deficiency

The development of hematopoietic stem cell B lymphocytes can be divided into different stages according to the sequential expression of proteins on the cell surface or within the cell, and the rearrangement of immunoglobulin (Ig) genes. Hematopoietic stem cells produce pluripotent progenitors (MPPs) and lymphoid-induced pluripotent progenitors (LMPPs) that lack the ability to self-renew. LMPPs have the ability to differentiate into common lymphoid progenitor cells (CLPs), granulocyte/macrophage progenitor cells (GMPs), or early T-cell progenitor cells (ETPs) [8]. The CLP compartment consists of all lymphoid progenitors (ALPs) and B-cell-biased lymphoid progenitors (BLPs). BLPs mainly differentiate into B-line cells [9].

The progression of HSCs in successive developmental stages requires changes in their cellular gene expression and chromatin status, reflecting genetic and epigenetic regulation [4]. In the context of sequential development, the core bypass network that orchestrates the fate specification of B cells is established. Transcription factors Ikaros, PU.1, and E2A regulate lymphocyte and myeloid fate selection, and EBF and Pax5 control B-cell specification and commitment. Ikaros transcription factor is a major regulator of the progression of HSC into the lymphatic system. Ikaros regulates cytokine receptor flt3, λ5 precursor B-cell receptor chain, and Rag1/2 gene expression, and its activity is required in LMPPs and early B-cell precursors. The expression of PU.1 in MPPs limits the fate of MEPs, and then the synergistic action of PU.1 and Ikaros-induced Gfi-1 (growth factory-independent-1) transcription factors establish selection of B-cell and myeloid fate by stabilizing PU.1 levels [10,11]. E2A promotes the production and/or maintenance of MPPs and LMPPs and is required for bone marrow restriction of LMPPs. In addition, E2A is required to promote progression of the B-cell lineage through a cascade of the regulatory factors EBF, Pax5, and Foxo1 [12,13]. Forced expression of EBF in MPPs activates the pedigree-related genes Pax5, λ5, VpreB, and Cd79b, and inhibits other fate-related genes, including c/EBPα [14]. Binding of the interferon regulatory factor IRF8 to the EBF promoter leads to transcriptional activation of EBF, while binding to the Sfp1 promoter inhibits PU.1 [15]. EBF in turn enhances E2A activity by inhibiting ID inhibitors of E2A [16].

The progression of BLPs to mature B cells involves multiple stages, with pro-B cells undergoing Rag-mediated assembly of the immunoglobulin heavy-chain (IgH) gene and successfully pairing IgH chain with alternative light-chain VpreB and λ5 to produce pre-B cells expressing pre-B cell receptors (pre-BCRs). This process allows the rearrangement of light-chain genes of immunoglobulin and successful pairing of IgH with IgK or IgX chains to produce immature B cells expressing BCRs that monitor the reactivity of B cells, which eventually differentiate into mature B cells from bone marrow [17].

B-cell deficiency (antibody deficiency disorder) is the most common type of immunodeficiency. It is caused by abnormal development and/or function of B cells and is the major primary immunodeficiency, accounting for approximately 50% of all PID diagnoses [18,19]. B cells develop in bone marrow and are the main cells of humoral immunity. The main functions of B cells are to produce antibodies, act as antigen-presenting cells, and secrete cytokines. Stimulated by antigens, B cells can differentiate into antibody-producing plasma cells and memory B cells to perform specific humoral immunity. A common feature of B-cell immunodeficiency disorders is a significant reduction in or absence of serum immunoglobulin. Antibody deficiency increases susceptibility to infection by bacterial pathogens, particularly Streptococcus pneumoniae and Hemophilus influenzae [20,21]. The manifestations and complications of B-cell developmental defects vary depending on the location or degree of functional impairment. There are three main types of primary B-cell defects: X-linked agammaglobulinemia [22,23,24,25,26,27,28,29], common variable immune deficiency [20,30,31,32,33,34,35,36,37,38,39,40], and high immunoglobulin syndrome [41,42,43,44].

X-linked agammaglobulinemia (XLA) was the first innate immune error found in humans and is the most common primary B-cell defect disease. It is characterized by B-cell and plasma cell defects and severe hypogammaglobulinemia; increased susceptibility to enveloped bacteria; and recurrent bacterial infections in infected men early in life [22]. The first case, an eight-year-old male child, was reported on by Ogden Bruton in 1952. The child experienced multiple bacterial pathogens: his serum sample was evaluated by protein electrophoresis and showed no globulin portion [23]. The affected gene, located on the long arm of the X chromosome, encodes a cytoplasmic tyrosine kinase named Bruton’s tyrosine kinase (BTK) [24,25]. Signals transduced with the help of BTK play a key role in the production of naive, mature B cells from the bone marrow into the circulation. When the expression level of BTK is low or gene mutation occurs, the developing B cells in bone marrow show maturation stagnation and cannot differentiate, and the level of mature B lymphocytes in the peripheral blood of patients is significantly reduced [26]. The clinical manifestations of XLA are repeated and severe bacterial infections can occur, such as upper respiratory tract infection, lower respiratory tract infection, nasal and pulmonary infection, otitis media, meningitis, osteomyelitis, sepsis, bronchitis, rheumatoid arthritis, etc. [27,28,29].

Common variable immune deficiency (CVID) is a major antibody deficiency and one of the most common primary immune deficiencies. In 1954, Sanford et al. reported the first clinical case of CVID in a 39-year-old woman with low serum levels of gamma globulin and recurrent infection [30]. In 1971, a committee of the World Health Organization coined the term “common variable immune deficiency” (CVID) to distinguish the less well-defined antibody deficiency syndrome from other conditions with more consistent clinical descriptions and Mendelian inheritance [31]. Patients with CVID, most of whom are diagnosed between the ages of 20 and 45 years, are characterized by significantly reduced serum immunoglobulin IgG and IgA, normal or low serum IgM, and defects in specific antibody production [20,32]. Different barriers to B-cell development occur in CVID, such as the failure of B cells to fully activate, proliferate normally, and eventually differentiate into plasma and/or memory B cells [33]. Patients with CVID often also have numerous T-cell abnormalities, such as defective T-cell activation [34], enhanced cell apoptosis [35], cytokine defects [36], lymphocytopenia [37], defects in mitogen and antigen proliferation [38], abnormal cell response to chemokines [39], etc. Clinical symptoms of CVID include severe lung disease, recurrent gastrointestinal infections, autoimmune, and inflammatory diseases [40].

High immunoglobulin M syndrome (HIGM), also known as immunoglobulin class switch recombination (Ig-CSR) deficiencies, is a rare primary immunodeficiency characterized by severely reduced serum levels of immunoglobulin A, G, and E; serum immunoglobulin M levels are normal or elevated [43]. The HIGM was first described by Rosen et al. in 1961. The HIGM was molecularly defined in a 1992 Notarangelo report on the CD40 ligand (CD40L) gene [41,42]. The HIGM phenotype has been observed in different single-gene immunodeficiency diseases, such as CD40L and CD40 defects, AICDA-encoded activation-induced cytidine deaminase (AID), and uracil-DNA glycosylase (UNG) deficiency disorders [43]. The CD40 molecule on B cells and its activated T cell ligand, CD40L, play a role in B-cell immunoglobulin isogenic signaling, and patients with CD40L mutations account for 65% of HIGM patients [44]. The main clinical symptoms of HIGM patients are upper and lower respiratory tract infections, otitis media, gastrointestinal infections, oral ulcers, autoimmune, lymphoid hyperplasia, and malignant tumors [42].

## 3. m^6^A Modification

More than 170 chemical modifications of RNA have been found in living organisms [45]. RNA modification has been found in all types of RNA molecules, including transfer RNA (tRNA), ribosomal RNA (rRNA), and messenger RNA (mRNA), as well as microRNA (miRNA), long non-coding RNA (lncRNA), and circRNA [45,46,47,48,49,50]. RNA modification plays an important role in RNA metabolism, including RNA structure formation; stability and dynamics [51]; RNA splicing; polyadenosine decomposition; transport; localization; and translatability. Currently, the most studied RNA modifications include *N*1-methyladenosine (m^1^A), 5-methylcytosine (m^5^C), *N*6-methyladenosine (m^6^A), *N*7-methylguanosine (m^7^G), *N*6,2′-*O*-dimethyladenosine (m^6^Am), 8-oxo-7,8-dihydroguanosine (8-oxoG), etc. [52,53,54,55].

Methylation of the N6 position of RNA (*N*6-methyladenosine [m^6^A]) is one of the most common post-transcriptional modifications of RNA and the most abundant internal mRNA modification. m^6^A plays an important role in almost every aspect of the mRNA life cycle, as well as in various cellular, developmental, and disease processes [56]. In mammalian cells, there are an average of 1–2 m^6^A sites per 1000 nucleotides [57,58]. m^6^A was first discovered in 1974 [59,60] and is mainly enriched in the 3′ untranslated region (3′ utrs), near the stop codon, inside and outside the long exon, intergenic region, intron, and 5′ untranslated region (5′ utrs) [61,62,63,64]. Similar to epigenetics, the deposition of RNA modification is dynamic and it has been identified that specific proteomes influence the fate of RNA, such as “writers” for catalytic modification deposition, “erasers” for catalytic modification removal, and “readers” for recognizing and binding modified nucleotides (Figure 2) [65]. The physiological roles of m^6^A and its reader in various biochemical processes have been studied and identified, such as embryonic stem cell differentiation [66], hematopoietic stem cell development [67,68,69], and immune response [70,71]. The characterization of these effector proteins in various biological systems underscores the multifaceted and adjustable nature of their function. Once the protein involved in m^6^A modification is abnormal, a series of diseases will be caused, including tumors, neurological diseases, embryonic development delay, etc.

### 3.1. m^6^A Writers

m^6^A writers are methyl transferases that catalyze the formation of m^6^A modification [72]. The multicomponent methyltransferase complex consists of S-adenosine methionine (SAM)-binding protein methyltransferase-like 3 (METTL3), methyltransferase-like 14 (METTL14) heterodimer catalytic cores, and various other methyltransferases [73,74]. The METTL3-METTL14 heterodimer is essential for the methylation process. METTL3 catalyzes the conversion of adenosine to m^6^A through its methyltransferase active domain, and METLL14 plays a key role in the substrate recognition process, providing structural support for METTL3 close to its active site, thus achieving catalysis [75,76]. The heterodimerization complex of the methyltransferase domain binds to the CCCH motif to form the minimum region required for the formation of m^6^A modification in vitro [75]. Wilms’ tumor 1-associated protein (WTAP) interacts with METTL3 and METTL14 to catalyze m^6^A methyltransferase activity in vivo. WTAP may also play a role in regulating the recruitment of m^6^A methyltransferase complex to mRNA targets [74,77]. Recent studies have shown that zinc finger protein Zc3h13 (Flacc) is required for the nuclear localization of the ZC3H13-WTAP-Virilizer-Hakai complex and promotes m^6^A methylation [78]. Through proteomic methods, KIAA1429 (also known as vir-like m^6^A methyltransferase associated (VIRMA)) is identified as another component of the m^6^A methyltransferase complex; KIAA1429 is one of the main interaction factors of WTAP [79,80,81]. In addition, RBM15 and its analog RBM15B are functional components of the methyltransferase complex and interact with METTL3 in a WTAP-dependent manner [82]. RBM15 and RBM15B bind to the uridine-rich region and then recruit the WTAP/METTL3 complex to methylate the nearby DRACH motif [82]. METTL16 has recently been identified as an m^6^A “writer” that plays a methyltransferase activity-dependent and independent role in gene regulation, promoting translation in an m^6^A independent manner [83]. METTL16-mediated methylation is mainly caused by small nuclear RNAs, some intron sites in pre-mRNA, and other ncRNAs [84,85,86].

### 3.2. m^6^A Erasers

m^6^A-modified deposition is reversible and dependent on demethylase. A-ketoglutarate-dependent dioxygenase alkB homology 5 (ALKBH5) and Fat Mass and Obesity Associated Protein (FTO) are “erasers” to reverse m^6^A methylation [87]. FTO is the first demethylated enzyme identified to catalyze the reversal of m^6^A methylation in mRNA, both in vitro and intracellularly [88,89]. In most cell lines, FTO is localized primarily in the nucleus and mediates 5–10% of total mRNA m^6^A demethylation. In leukemia cells, FTO is highly abundant in the cytoplasm, mediating up to about 40% of m^6^A demethylation [65]. In addition, AlkB homolog 3 (ALKBH3) was found to preferentially act on m^6^A modifications in tRNAs [90]. Because m^6^A demethylase is distributed differently in tissues and plays an important role in regulating m^6^A methylation, additional cell or tissue-specific demethylases may exist to act on different RNA substrates [91].

### 3.3. m^6^A Readers

m^6^A modification sites can be recognized by “reader” proteins to regulate RNA metabolism, splicing, translocation, degradation, and processing [91]. Some m^6^A binding proteins with YTH domains, including YTHDF1, YTHDF2, YTHDF3, YTHDC1, and YTHDC2, act as “readers” of m^6^A to regulate the translation and mediated degradation of m^6^A-modified RNA [56,92]. YTHDF1 can enhance mRNA translation, YTHDF2 can promote mRNA degradation, and YTHDF3 can enhance both translation and degradation. The main function of YTHDFs is to inhibit gene expression by enhancing the degradation of methylated mRNA in cytoplasm [76,93,94,95]. YTHDC1 binds to certain m^6^A sites in both mRNA and non-coding RNA, while YTHDC2 mainly binds to non-coding RNA [96,97]. YTHDC1 of Drosophila melanogaster participates in sex determination and dose compensation by regulating selective splicing of Sxl. In humans, YTHDC1 also plays a role in dose compensation. YTHDC1 interacts with splicing factors to regulate alternative splicing and nuclear output. YTHDC2 is a nucleoplasmic protein that only exists in mammals. It is characterized by a gyrase domain, anchor repeat sequence, YTH domain, and DUF1065 domain [98]. Later, other readers were discovered: Eukaryotic translation initiation factor 3 (EIF3), heterogeneous nuclear ribonucleoprotein (hnRNPC and hnRNPA2/B1), insulin-like growth factors (IGF2BP1, IGF2BP2, and IGF2BP3), proline-rich and curled protein 2A (PRRC2A), and fragile X mental retardation protein (FMRP), etc. YTHDF1 binds to the m^6^A site around the stop codon of mRNA and can recruit the 40S ribosomal complex, including eIF3, eukaryotic translation initiation factor 4E (eIF4E), eukaryotic translation initiation factor 4G (eIF4G), poly (A)-binding protein (PABP), and 40S ribosome subunits to promote the translation of target RNA [95]. The eIF3 can be recruited directly by m^6^A in the 5′ UTR region of the transcript and then recruited into the 43S ribosomal pre-initiation complex, promoting cap-independent translation [64]. hnRNPA2/B1 can recognize m^6^A on the transcriptional subset of primary microRNA (pri-miRNA) and interact with the microRNA microprocessor complex protein DGCR8 to promote the processing of pri-miRNA [46]. IGF2BPs can recognize m^6^A, promote mRNA stability and translation, and depend on m^6^A [99]. Deficiency of PRRC2A, a novel m^6^A-specific binding protein found in nerve cells, leads to hypomyelination by affecting oligodendrocyte regulation in the brain [100]. By studying the regulatory mechanism of RNA-binding protein FMR1, we found that FMR1 is a novel m^6^A reader, which affects the translation of target mRNA and the transport of mRNA particles [101].

## 4. m^6^A Modifications in B-Cell Development and B-Cell-Related Diseases

m^6^A modification and its regulatory factors regulate the expression of genes, which are associated with many B-cell diseases (Table 1).

### 4.1. The role of m^6^A Writers in the Development and Function of B Cells

m^6^A modification can regulate the development of early B cells. Deletion of METTL14 significantly reduced m^6^A methylation in developing B cells and severely hindered the development of mouse B cells. The large-pre-B-to-small-pre-B transition process in METTL14 knockout mice was impaired. Loss of METTL14 in developing B cells reduces YTHDF2 binding to its target and specifically leads to up-regulation of a set of YTHDF2-bound transcripts. YTHDF2-mediated degradation of mRNA is key to the transition from pro-B stage to large pre-B stage, and both METTL14 deletion and YTHDF2 deletion significantly block IL-7-induced pro-B-cell proliferation [105].

Grenov et al. have shown in their studies that Mettl3 regulates the response of GC B cells through YthDF2-mediated degradation of genes associated with oxidative phosphorylation and IGF2BP3, enhancing the stability of m^6^A-modified Myc transcripts. METTL3 deletion in GC B cells slowed down the cell cycle process and reduced the expression of genes related to proliferation and oxidative phosphorylation. m^6^A interaction factor IGF2BP3 is required for GC persistence to support Myc transcriptional stabilization and downstream pathways. YTHDF2 as a reader of m^6^A can regulate appropriate gene expression and function of mitochondrial respiration [102]. In Huang et al.’s study, it was found that METTL14-mediated RNA modification of m^6^A is essential for germinal center (GC) B-cell response in mice. When METTL14 was specifically deleted from B cells, the response of GC B cells was impaired, and BCR and CD40 signals in GC B cells were attenuated. METTL14-mediated m^6^A indirectly up-regulates the expression of genes critical for positive selection and proliferation of GC B cells by promoting mRNA decay of genes encoding a set of negative immunomodulators, including Lax1 and Tipe2 [106]. The deletion of METTL3 affected the stability of Myc mRNA, while the deletion of METTL14 did not reduce the level of Myc mRNA in GC B cells. This difference may be due to incomplete functional overlap between METTL3 and METTL14, with METTL3 having methyltransferase activity instead of METTL14 [119,120].

In Jiang et al.’s study, genetic analysis of the B-cell CRISPR/Cas9 system was used to identify positive and negative regulators of CD40 response. In the study, WTAP components VIRMA/KIAA1429 had a strong negative regulatory effect on CD40, and WTAP regulated CD40 response by negatively regulating CD40 mRNA levels [107]. GC is a secondary lymphoid organ structure essential for key aspects of B-cell development, differentiation, somatic super-mutation, and class transformation recombination. Interference with CD40/CD40L signaling collapses GC, which is the basis of adaptive humoral immune response [121,122].

In the study of Xu et al., it was found that the expression levels of ZC3H13, RBM15, RBM15B and VIRMA were positively correlated with the expression of RAB39B through comprehensive biological information analysis. RAB39B is associated with proliferation, apoptosis and drug sensitivity of diffuse large B-cell lymphoma (DLBCL), the most common aggressive lymphoma. RAB39B can be used as an effective biomarker for the diagnosis and treatment of DLBCL [108]. In a study by Raffel et al., loss of RBM15 resulted in the obstruction of pro/pre-B-cell differentiation and the loss of peripheral B cells in adult mice. It has also been shown that RBM15 is essential for B lymphocyte generation and has inhibitory effects on myeloid, megakaryocytes, and the progenitor cell compartment [109]. Niu et al. found that RBM15 may function in part by regulating the expression of the proto-oncogene c-Myc, which is necessary for normal hematopoietic stem cell-niche interaction and normal promotion of adult hematopoietic cells and normal megakaryocyte development [110].

### 4.2. The Role of m^6^A Erasers in B Cell Development and Function

m^6^A methylation can significantly improve the expression of innate immune cells associated with inflammatory processes [123]. FTO is a potential anti-inflammatory target [111]. The expression of RAB39B, an effective biomarker of DLBCL, was significantly positively correlated with FTO and ALKBH5 [108]. Translation-regulated lncRNA1 (TRERNA1) was first reported as an enhancer-like RNA, which can mediate the expression of its neighboring genes [124]. TRERNA1 was positively correlated with lymph node metastasis, and its expression stimulated the invasion and metastasis of breast cancer and gastric cancer [125,126]. TRERNA1 modifies its promoter region by H3K27me3 and recruits EZH2 to silence the expression of cyclin-dependent kinase inhibitor p21 in an epigenetic manner. TRERNA1 can be modified by ALKBH5, the up-regulation of ALKBH5 promotes the expression of TRERNA1, and the N6-methyladenosine-modified TRERNA1 mediated by ALKBH5 promotes the occurrence of DLBCL [112]. In addition, ALKBH5 is related to the growth of Myc-dysregulated B-cell lymphoma, and inhibition of ALKBH5 can effectively inhibit the growth of MYC-dysregulated B-cell lymphoma, both in vitro and in vivo. Myc activated the expression of ALKBH5 and decreased the level of m^6^A in mRNA [113].

### 4.3. The role of m^6^A Readers in the Development and Function of B Cells

Jiang et al. showed that m^6^A reader YTHDF2 was a negative regulator of CD40, and the knocking out of YTHDF2 could increase the abundance of CD40 [107]. The expression levels of YTHDC1, YTHDC2, YTHDF1, YTHDF2, and YTHDF3 in RAB39B high-expression cells were significantly up-regulated [108]. Grenov et al. reported a post-transcriptional mechanism that inhibits plasmoblastic genetic programming and promotes GC B cells. In their study, using single-cell RNA sequencing (RNA-seq) techniques and transgenic mice, they found that antigen-specific B-cell precursors up-regulate YTHDF2 in the pre-GC phase, thereby enhancing the decay of methylated transcribed proteins during the early stage of B-cell immune response [114]. Recent studies have shown that hnRNPC is closely associated with alternative splicing associated with overall survival in DLBCL [127]. Yin et al. found that hnRNPA2/B1 was associated with the proliferation of human glioma cells. The down-regulation of hnRNPA2/B1 led to the inactivation of AKT and STAT3 signaling pathways, and ultimately decreased the expression of B-cell lymphoma-2 (Bcl-2), cyclin D1 and proliferating cell nuclear antigen (PCNA) [117]. In addition, IGF2BP1-3 has been shown to play a role in B-lymphoid precursor tumors [118]. In B-cell acute lymphoblastic leukemia (B-ALL), the high mRNA expression of IGF2BP3 is associated with the high expression of proliferative “metagene” markers and CDK6 [128]. IGF2BP3 is also used as a diagnostic and prognostic marker for several malignant tumors. Mutations in the RRRC2A gene affect the risk of non-Hodgkin lymphoma (NHL) [115].

## 5. Conclusions

The formation and function of B lymphocytes largely depends on the precise regulation of multilayer gene expression. More and more evidence is showing that post-transcriptional modification of RNA is another important regulatory link of gene expression, which can regulate mRNA degradation, splicing or translation during B-lymphocyte generation. This review briefly introduces the development and maturation of B cells. Many studies have shown that B-cell-related diseases are related to m^6^A modifier regulators. The identification of disease-causing genes and modification factors may help clarify the regulatory requirements for normal B-cell development as well as the potential basis for some common diseases and the search for new drug targets. In short, this topic has high research value and needs further study.

## Figures and Tables

**Figure 1 ijms-24-04721-f001:**
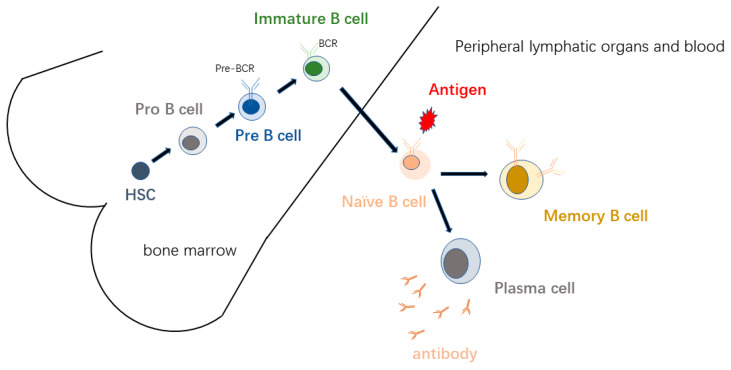
The development of B cells. B-cells develop in the bone marrow (BM) and later mature in the secondary lymphoid organs.

**Figure 2 ijms-24-04721-f002:**
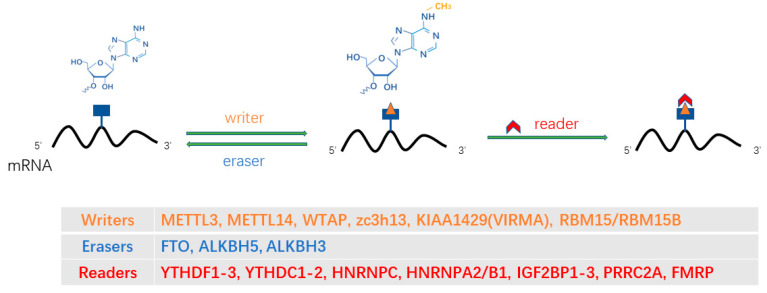
m^6^A modification and related proteins. “Writers” for catalytic modification deposition, “erasers” for catalytic modification removal, and “readers” for recognizing and binding modified nucleotides.

**Table 1 ijms-24-04721-t001:** m^6^A regulatory factors and roles in the development and functioning of B cells.

m^6^A Regulatory Factor		The Role in the Development and Functioning of B Cells	Refs
Writers	METTL3	B-cell development	[102,103,104]
	METTL14	B-cell development	[105,106]
	WTAP	Regulating CD40 response is related to B-cell development	[107]
	Zc3h13	Related to DLBCL	[108]
	RBM15/RBM15B	Associate with DLBCL, hematopoietic and normal cell development	[108,109,110]
Erasers	FTO	Related to DLBCL	[108,111]
	ALKBH5	Related to DLBCL	[112,113]
Readers	YTHDF1-3	Related to DLBCL, YTDHF2 is associated with B-cell development and is a negative regulator of CD40	[107,108,114,115]
	YTHDC1-2	Related to DLBCL	[108]
	hnRNPC	Related to DLBCL	[116]
	hnRNPA2/B1	Unknown	
	IGF2BP1-3	Associated with B-lymphoid precursor tumors	[117]
	PRRC2A	Associated with NHL	[118]

## Data Availability

Not applicable.

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
