# Peer review of "The Role of m6A Modifications in B-Cell Development and B-Cell-Related Diseases"

_ijms, 2023, doi:10.3390/ijms24054721_

Round 1

Reviewer 1 Report

Introduction: Worth specifying the difference between primary and secondary IDs – i.e congenital and acquired, although the review focuses on PIDs so this section could be streamlined

Line 35: PID rather than Pid

Line 40: Do the authors mean “B cells mature in the bone marrow”?

Line 46: C. perfringens? Or bacterial pathogens?

Line 53-57: sentences are unclear (also repeated in Fig 1 legend). Without a brief discussion of the major molecular regulators of these developmental processes, this section appears vague

Fig 1 – a graphical summary of the genetic causes of PIDs described could be more appropriate

Line 71: agammaglobulinemia, rather than aglobulinemia. Similarly, line 77 – no gamma globulin band

Line 106: HIGM – High Immunoglobulin M syndrome

Line 114: T cell ligand is CD40L, rather than CD40

Line 140: should refer to Fig 2

Line 174: METTL16-mediated methylation? Or is it methylated itself?

Section 3.3: The YTHDF proteins are discussed in terms of function, while YTHDCs only in terms of binding. This section could be improved by clearly highlighting major functional groups of m6A readers

Table 2: This table would be useful but a further column including relevant references would be critical

Line 230: degradation of mRNA.

Line 232: deletion in GC B cells

Line 264: RBM15 function or recruitment is mediated by c-Myc?

Line 268: Do the authors refers to expression of inflammatory genes, rather than cells?

Line 271: would benefit from highlighting the significance of TRERNA1 in DLBCL

Line 287-294: This section appears disjointed and would benefit from clearly highlighting B cell relevant functions of m6A readers

Section 4 (particularly 4.2 and 4.3) are titled B cell development and function but mainly discuss B cell malignancy.

Finally, there are a number of minor errors around punctuation, capitalization and grammar, sentences ending in “etc.”, and murine vs human nomenclature.

Overall, the review is of interest to a broad audience and highlights a number of major recent developments in the field. However, it reads as 2 separate sections – a basic review of B cell-related PIDs, and a review on roles of m6A modification in B cell development/function – with no attempt to bring these together. In my opinion, this distracts from the main message and the review would be improved by removing the PID section and clearly introducing relevant B cell activation and signaling pathways before discussing how these are modulated by m6A writers, readers and erasers.

Author Response

Dear Editor,

Thank you very much for your kind information regarding your comments and the reviewer’s criticisms for our manuscript entitled “Common B-cell immunodeficiency and the role of m6A modi-fications in B-cell development and B-cell-related diseases” . Those comments are all very valuable and very helpful revising and improving our paper, as well as the important guiding significance to our research. We have made a detailed reversion on the manuscript according to the editor’s criticisms and suggestions, and resubmit the revised manuscript for your consideration of publication in the  Molecular Sciences, which we hope meet with approval.

Thank you again for your time and consideration.

The main corrections in the manuscript and the responds to the reviewer’s comments are as following:

  1. Introduction: Worth specifying the difference between primary and secondary IDs – i.e congenital and acquired, although the review focuses on PIDs so this section could be streamlined

Response: Thanks for your advice. I've simplified this part.

  1. Line 35: PID rather than Pid

Response: Thanks to reviewer for your careful review, we have corrected it.

  1. Line 40: Do the authors mean “B cells mature in the bone marrow”?

Response: Thanks to reviewer for your review. There is a mistake in this sentence. I've changed to B cells developing in the bone marrow.

  1. Line 46: C. perfringens? Or bacterial pathogens?

Response: Thanks for your careful review. Those are bacterial pathogens.

  1. Line 53-57: sentences are unclear (also repeated in Fig 1 legend). Without a brief discussion of the major molecular regulators of these developmental processes, this section appears vague

Response: Thanks to reviewer for your careful review, we have added a part of content to explain later.

  1. Fig 1 – a graphical summary of the genetic causes of PIDs described could be more appropriate

Response: Thanks for your advice.

  1. Line 71: agammaglobulinemia, rather than aglobulinemia. Similarly, line 77 – no gamma globulin band

Response: Thanks to reviewer for your careful review, we have corrected it.

  1. Line 106: HIGM – High Immunoglobulin M syndrome

Response: Thanks to reviewer for your careful review, we have corrected it.

  1. Line 114: T cell ligand is CD40L, rather than CD40

Response: Thanks for your review. We have corrected it.

  • Line 140: should refer to Fig 2

Response: Thanks to reviewer for your careful review, we have corrected it.

  • Line 174: METTL16-mediated methylation? Or is it methylated itself?

Response: Thanks for your careful review. This should be MettL16-mediated methylation. We have corrected it.

  • Section 3.3: The YTHDF proteins are discussed in terms of function, while YTHDCs only in terms of binding. This section could be improved by clearly highlighting major functional groups of m6A readers

Response: Thanks to reviewer for your careful review, we have supplemented the functionality of YTHDCs. line 178-182.

  • Table 2: This table would be useful but a further column including relevant references would be critical

Response: Thanks for your review. We have included references in the table.

  • Line 230: degradation of mRNA.

Response: Thanks for your review. We have corrected it.

  • Line 232: deletion in GC B cells

Response: Thanks for your review. We have corrected it.

  • Line 264: RBM15 function or recruitment is mediated by c-Myc?

Response: Thanks to reviewer for your review. There is a mistake in this sentence. We have corrected it into “RBM15 may function in part by regulating the expression of the proto-oncogene c-Myc”. line 249.

  • Line 268: Do the authors refers to expression of inflammatory genes, rather than cells?

Response: Thanks for your review. We're referring to cells.

  • Line 271: would benefit from highlighting the significance of TRERNA1 in DLBCL

Response: Thanks to reviewer for your careful review, we have added it. line 257-263.

  • Line 287-294: This section appears disjointed and would benefit from clearly highlighting B cell relevant functions of m6A readers

Response: Thanks for your review. We've added a little bit here to make it look more coherent. line 368.

  • Section 4 (particularly 4.2 and 4.3) are titled B cell development and function but mainly discuss B cell malignancy.

Response: Thanks to reviewer for your review. Due to the lack of research in this area, the literature we can find is mostly about this aspect, so there are more discussions on B-cell malignant tumors.

Reviewer 2 Report

This is a review paper that aims to integrate RNA modification (especially m6A modification), B cell development, and B cell immunodeficiency. The topic itself is original, but the manuscript still needs major improvements.

Major comments:

1. There is only a short discussion integrating RNA modification and immunodeficiency. The general focus is rather on the mechanisms of RNA modification, with some correlations with B cell development. The review might benefit from more elaborate future perspectives, e.g what genes to study in details to prove the roles of RNA modification in B cell immunodeficiency, and what methods to employ.

2. Correlation with malignancy (e.g lymphoma) might be avoided, except when the malignancy is indeed disproportionately present in B cell immunodeficient patients. Otherwise, the review should not limit itself to B cell immunodeficiency, and the introduction should then include malignancy.

Minor comments:

1. The introduction does not present the relevance of RNA modification in immunodeficiency.

2. Please review the clinical terms used throughout the review article. E.g HIGM (Hyper IgM syndrome), XLA (X-linked agammaglobulinemia). NHL was at least once referred to as "Fehodgkin's" (line 294).

3. Avoid unnecessary ambiguity, e.g "some" ("some leukemia cells", line 182).

4. Please review pathogens which were mentioned, e.g "bacteria perfringens" (line 46), most probably a mistake.

5. Table 2, "glioma" was mentioned in the column about B cells.

6. Pay closer attention to references of surnames, e.g 'Amalie' (line 229) refers to 'AC Grenov' (reference 97) and should consequently read 'Grenov'.

Author Response

Dear Editor,

Thank you very much for your kind information regarding your comments and the reviewer’s criticisms for our manuscript entitled “Common B-cell immunodeficiency and the role of m6A modi-fications in B-cell development and B-cell-related diseases” . Those comments are all very valuable and very helpful revising and improving our paper, as well as the important guiding significance to our research. We have made a detailed reversion on the manuscript according to the editor’s criticisms and suggestions, and resubmit the revised manuscript for your consideration of publication in the  Molecular Sciences, which we hope meet with approval.

Thank you again for your time and consideration.

There is only a short discussion integrating RNA modification and immunodeficiency. The general focus is rather on the mechanisms of RNA modification, with some correlations with B cell development. The review might benefit from more elaborate future perspectives, e.g what genes to study in details to prove the roles of RNA modification in B cell immunodeficiency, and what methods to employ.

Response: Thanks to reviewer for your careful review, We have highlighted some of the genes involved in this manuscript. Line 257, 285.

Correlation with malignancy (e.g lymphoma) might be avoided, except when the malignancy is indeed disproportionately present in B cell immunodeficient patients. Otherwise, the review should not limit itself to B cell immunodeficiency, and the introduction should then include malignancy.

Response: Thanks to reviewer for your careful review. Because there are too few studies on RNA modification related to B cells, most of the literatures we can find are related to malignant tumors, so there are too many descriptions in this aspect. We also hope that some researchers can get ideas from this paper to study the role of RNA modification in the development and function of B cells.

The introduction does not present the relevance of RNA modification in immunodeficiency.

Response: Thanks for your review. We edited out the immune deficiency part of the manuscript.

Please review the clinical terms used throughout the review article. E.g HIGM (Hyper IgM syndrome), XLA (X-linked agammaglobulinemia). NHL was at least once referred to as "Fehodgkin's" (line 294).

Response: Thanks for your careful review. We have checked and corrected it.

Avoid unnecessary ambiguity, e.g "some" ("some leukemia cells", line 182).

Response: Thanks to reviewer for your careful review. We have corrected it.

Please review pathogens which were mentioned, e.g "bacteria perfringens" (line 46), most probably a mistake.

Response: Thanks for your careful review. We have checked and corrected it.

Table 2, "glioma" was mentioned in the column about B cells.

Response: Thanks for your careful review. We have checked and corrected it.

Pay closer attention to references of surnames, e.g 'Amalie' (line 229) refers to 'AC Grenov' (reference 97) and should consequently read 'Grenov'.

Response: Thanks for your careful review. We have checked and corrected it.

In all, the comments from editor and reviewers are quite helpful to our manuscript, and we have revised our manuscript point by point. Once again, thank you very much for your good comments and suggestions.

Round 2

Reviewer 2 Report

The manuscript was much improved and now reads much better. There might not be any point of keeping Table 1, as immunodeficiency is not a major part of the manuscript as a whole.

Minor remark:

The last sentence (line 299-301, "But the role of m6A modification in B cell development and B-cell related diseases is of high research value and more research needs to be research.") is too long, redundant, and could be formulated better.

Author Response

Dear Editor,

Thank you for your decision and constructive comments on my manuscript entitled “The role of m6A modifications in B-cell development and B-cell-related diseases”. We have carefully considered the suggestion of Reviewer and make some changes. We have tried our best to improve and made some changes in the manuscript.

Thank you again for your time and consideration.

Best regards,

Shu-qi Wang

Beijing Key Laboratory for Animal Genetic Improvement, National Engineering Laboratory for Animal Breeding, Key Laboratory of Animal Genetics and Breeding of the Ministry of Agriculture, College of Animal Science and Technology, China Agricultural University.

Table 1 has been deleted.  The last sentence statement has been modified